# Hyperpolarized ^13^C-Pyruvate to Assess Response to Anti-PD1 Immune Checkpoint Inhibition in YUMMER 1.7 Melanoma Xenografts

**DOI:** 10.3390/ijms24032499

**Published:** 2023-01-28

**Authors:** Chantale Farah, Marie-Aline Neveu, Caroline Bouzin, Zorica Knezevic, Bernard Gallez, Eleonora Leucci, Jean-François Baurain, Lionel Mignion, Bénédicte F. Jordan

**Affiliations:** 1Biomedical Magnetic Resonance Research Group, Louvain Drug Research Institute, Université Catholique de Louvain (UCLouvain), B-1200 Brussels, Belgium; 2Laboratory of Tumor Inflammation and Angiogenesis, Department of Oncology, K.U. Leuven, B-3001 Leuven, Belgium; 3IREC Imaging Platform, Institut de Recherche Expérimentale et Clinique, Université Catholique de Louvai, (UCLouvain), B-1200 Brussels, Belgium; 4Laboratory for RNA Cancer Biology, Department of Oncology, K.U. Leuven, B-3001 Leuven, Belgium; 5Nuclear and Electron Spin Technologies (NEST) Platform, Louvain Drug Research Institute (LDRI), Université Catholique de Louvain (U.C. Louvain), B-1200 Brussels, Belgium; 6Molecular Imaging and Radiation Oncology (MIRO) Group, Institute de Recherche Expérimentale et Clinique (IREC), B-1200 Brussels, Belgium

**Keywords:** melanoma, tumor metabolism, immunotherapy, response biomarkers, ^13^C-MRS

## Abstract

There is currently no consensus to determine which advanced melanoma patients will benefit from immunotherapy, highlighting the critical need to identify early-response biomarkers to immune checkpoint inhibitors. The aim of this work was to evaluate in vivo metabolic spectroscopy using hyperpolarized (HP) ^13^C-pyruvate and ^13^C-glucose to assess early response to anti-PD1 therapy in the YUMMER1.7 syngeneic melanoma model. The xenografts showed a significant tumor growth delay when treated with two cycles of an anti-PD1 antibody compared to an isotype control antibody. ^13^C-MRS was performed in vivo after the injection of hyperpolarized ^13^C-pyruvate, at baseline and after one cycle of immunotherapy, to evaluate early dynamic changes in ^13^C-pyruvate–^13^C-lactate exchange. Furthermore, ex vivo ^13^C-MRS metabolic tracing experiments were performed after U-^13^C-glucose injection following one cycle of immunotherapy. A significant decrease in the ratio of HP ^13^C-lactate to ^13^C-pyruvate was observed in vivo in comparison with the isotype control group, while there was a lack of change in the levels of ^13^C lactate and ^13^C alanine issued from ^13^C glucose infusion, following ex vivo assessment on resected tumors. Thus, these results suggest that hyperpolarized ^13^C-pyruvate could be used to assess early response to immune checkpoint inhibitors in melanoma patients.

## 1. Introduction

Melanoma is a skin cancer caused by the uncontrolled growth of abnormal melanocytes. It is known for its poor survival rates in its advanced form and increased incidence rates over the last decades [1]. Indeed, early stages of melanoma are curable after surgery; meanwhile, in metastatic diseases, less than 10% of patients will recover [2]. The introduction of targeted therapies in 2012 led to unprecedented progress in melanoma care; however, the benefits are often transient due to the development of resistance [3]. Interestingly, immunotherapies based on immune checkpoint inhibitors do offer a long-term effect with durable clinical benefits [4,5], yet a significant proportion of patients display intrinsic resistance to this therapy. A response rate of 59% was observed following treatment with combined immune checkpoint inhibitors [6]. Immune checkpoints are receptors expressed on the membrane of activated T cells, and control the activation or inhibition of the immune responses [7,8]. Immune checkpoint blockades use antibodies to inhibit the interaction between receptors expressed at the surface of activated T cells and on cancer cells to unleash immune responses [7,8,9]. T cells play a crucial role in anti-tumor immune defense; they become activated after recognizing tumor antigens and consequently migrate to eliminate cancer cells. This activation is affected by the tumor microenvironment (TME), i.e., the interaction of the stimulatory and inhibitory ligand receptors between T cells, dendritic cells, and macrophages, with the tumor cells acting as immunosuppressors [10,11].

Immunotherapy using immune checkpoint inhibitors (ICIs) constituted a major breakthrough in the treatment of cancer over the past several years [12]. Ipilimumab was the first FDA-approved antibody in melanoma that inhibits signaling via cytotoxic T-lymphocyte-associated proteins (CTLA-4) on T cells. Three years later, an anti-programmed death-1 (anti-PD1) monoclonal antibody, pembrolizumab, was also approved for the treatment of patients with metastatic melanoma [13]. In the same year, an anti-programmed cell death protein 1 (anti-PD1), nivolumab, gained approval [14]. These immunotherapies showed clinical responses that last for several years. However, primary resistance mechanisms are still present [15]. Despite the fact that immunotherapies have displayed long-term effects with durable benefits on a subset of patients, there are no tools which can identify those patients that will benefit from immunotherapies [4,5]. The identification of robust biomarkers remains a challenge in the assessment of drug resistance. Several biomarkers have been studied, including PD-L1 expression, mismatch repair deficiency, and tumor-infiltrating lymphocytes (TILs); however, unfortunately, these have failed to show predictive value for treatment responses in the clinical setting [16].

Recently, a metabolic competition between tumor cells and immune cells has been reported. In 2015, different works reported that the aerobic glycolysis (the Warburg effect) promotes the depletion of extracellular glucose, thereby restricting glucose to T cells and rendering T cells dysfunctional [17,18]. Moreover, it has been shown that the PD1 blockade leads to decreased glycolysis in tumors [17]. Besides metabolic competition, tumor cells create an environment characterized by hypoxia, low pH, and the presence of lactic acid, which all play an immune inhibitory role [19,20]. In this context, it is relevant to assess treatment responses to cancer immunotherapy by evaluating potential shifts in the tumor metabolism. 

Among several non-invasive molecular imaging techniques currently used in the preclinical and clinical settings, ^18^F-fluoro-deoxy-glucose-positron emission tomography (FDG-PET) was suggested to be useful as a marker of melanoma response [21]. Early FDG-PET scans at day 15 in patients treated with targeted therapy (dabrafenib) on phase I trials demonstrated early, complete, or partial metabolic responses, potentially with prognostic significance [22,23]. In the scope of immunotherapy, few studies assessed ^18^F-FDG PET/CT as an early predictor of responses to PD-1 or CTLA-4 blockades in patients with non-small-cell lung carcinoma or melanoma [24,25]. In particular, ^18^F-FDG PET/CT performed after two cycles of ipilimumab was shown to be highly predictive of the final treatment outcome in patients with progressive and stable metabolic disease in metastatic melanoma [26]. However, the distinction of patients between pseudoprogression and progressive disease has been shown in several consecutive studies to be limited using ^18^F-FDG PET/CT [27,28]. More recently, one clinical study highlighted that early scans performed from three to four weeks after the start of ICI treatment could be predictive of the response in patients with advanced melanoma when functional and anatomic imaging parameters were combined, further stressing the need to conclusively develop larger cohorts [24].

There is, therefore, currently a lack of robust noninvasive imaging markers to evaluate early responses to immunotherapy in the clinical setting. The purpose of this study was to investigate the ability of non-invasive metabolic imaging to evaluate responses to immune checkpoint blockades with anti-PD1 using ^13^C-MRS (magnetic resonance spectroscopy) metabolic imaging. 

^13^C-metabolic imaging is a cutting-edge technology that is now considered to assess sensitivity to therapy in the preclinical setting as well as in clinical studies, with a higher specificity than ^18^F-FDG, by monitoring the fate of glucose beyond glucose uptake via tumor cells [29,30]. The monitoring of ^13^C-enriched metabolic substrates can be monitored using tools such as ^13^C magnetic resonance spectroscopy (MRS) or imaging (MRI). MRS can investigate static metabolic processes in vivo after injecting a ^13^C-enriched substrate (i.e., ^13^C-glucose or ^13^C-glutamine) [31]. Moreover, dynamic metabolic processes can be assessed by temporarily boosting the ^13^C NMR signal of some key metabolic substrates (such as ^13^C-pyruvate) using hyperpolarization, thereby allowing for the dynamic measurement of real-time metabolic conversions in vivo [32]. Notably, dynamic nuclear polarization (DNP) can increase ^13^C-MRS sensitivity 10,000-fold, providing an opportunity to detect real-time metabolic fluxes in vivo, such as ^13^C-pyruvate–^13^C-lactate exchange, in preclinical studies as well as in patients [33,34,35,36].

In this study, we characterized the ^13^C-metabolic profile in response to the PD1 blockade in YUMMER1.7 syngeneic melanoma xenografts characterized for BrafV600E/wt Pten−/−Cdkn2−/−somatic mutations [37], allowing for the use of immunocompetent mice to fully integrate all aspects of the tumor microenvironment. We aimed to evaluate the relevance of hyperpolarized (HP) ^13^C-pyruvate and ^13^C-MRS fluxomic after [U-^13^C]-glucose injection as potential response markers to immunotherapy in the preclinical setting.

## 2. Results

### 2.1. Anti-PD1 Monoclonal Antibody Delays Tumor Growth in Syngeneic YUMMER1.7 Melanoma Xenografts

An anti-PD1 monoclonal antibody was used to evaluate the therapeutic effects of the immune checkpoint blockade in vivo. C57BL/6 mice were subcutaneously inoculated with YUMMER1.7 (1.5 × 10^6^ cells) into the right leg and treated with isotype or anti-PD1 antibodies via intraperitoneal injection 3 times per week for 2 weeks (2 cycles of immunotherapy) (Figure 1A).

Anti-PD1 significantly delayed (*p* = 0.04) the progression of YUMMER1.7 melanoma xenografts compared to the isotype group, with a significant difference between groups upon day 9 after treatment initiation, as shown in Figure 1B,C. The growth delay was potentially induced by a combined effect of anti-PD1 treatment on tumor cell death and tumor cell proliferation, as shown by the trends observed with IHC of KI-67 and caspase-3, respectively, assessed after two cycles of immunotherapy (Figure 1D,E). At the same timepoint, the infiltration of T cells and macrophages showed an increasing trend in the anti-PD1 group at the edge of the tumors (Figure 1F). Of note, there was a lack of change in T cell infiltration and in the number of macrophages in the central regions of the tumors. 

### 2.2. Hyperpolarized ^13^C-Pyruvate Detects Real-Time Metabolic Changes Induced by Anti-PD1 In Vivo

In order to determine metabolic differences between control and treated mice, we performed intravenous injections of HP [1-^13^C] pyruvate to mice bearing YUMMER1.7 xenografts to monitor, in vivo, the real-time exchange between [1-^13^C] pyruvate and [1-^13^C] lactate using in vivo ^13^C-MRS (Bruker Biospec, 11.7T). Figure 2B shows representative evolutions of ^13^C pyruvate and ^13^C lactate peaks over time, from which the area under the curve and ratios were calculated. Anti-PD1-treated tumors showed a significant decrease in the ^13^C lactate over the total ^13^C signal (LA/^13^C) ratio at day 5 compared to day 0, whereas the isotype control tumors did not show any significant change over time (Figure 2C,D). The anti-PD1-treated tumors also displayed a significant increase in the ^13^C pyruvate over total ^13^C signals (PA/^13^C) (Figure 2E,F). Consequently, a significant decrease was observed in the evolution of the ratio of hyperpolarized (HP) ^13^C lactate to ^13^C pyruvate (LA/PA) after treatment with anti-PD1 (Figure 2G). These data point out that HP ^13^C-pyruvate administered after one cycle of anti-PD1 treatment was able to detect metabolic changes before any significant change in tumor size could be observed.

### 2.3. Ex Vivo Metabolic Profiling Using ^13^C-Glucose Tracing Is Not Modified in Response to Anti-PD1 Treatment

To further characterize the glucose metabolism in the YUMMER1.7 model in response to anti-PD1 and isotype antibodies, we conducted fluxomic experiments after injecting uniformly-labeled U-^13^C glucose at day 6 on the same cohort. The downstream metabolites were detected after tumor resection (75 min after U-^13^C glucose infusion) and metabolite extraction. Typical spectra, acquired using high-resolution ^13^C-NMR (Bruker Ascend, 600 MHz), are shown in Figure 3A,B. 

We did not observe any significant change in the steady-state levels of ^13^C-MRS ex vivo detectable metabolites in YUMMER1.7 xenografts, i.e., ^13^C-lactate, ^13^C-alanine, and ^13^C-glutamate, between the isotype and anti-PD1-treated tumors (Figure 3C). 

### 2.4. Expression of Metabolic Transporters and LDH-A in Response to Anti-PD1

Since we observed a lower HP ^13^C lactate to ^13^C pyruvate ratio in response to anti-PD1 treatment, and since this ratio has been shown to be dependent on the LDH-A activity, as well as on the activity of the monocarboxylate transporters MCT-1 and MCT-4, and the glucose transporter GLUT-1, we assessed their expression via IHC after one cycle of anti-PD1 treatment to match the timing of the metabolic imaging experiments. A lack of change in GLUT-1 staining was observed in response to anti-PD1 treatment (Figure 4A), yet GLUT-4 was not assessed in this study. Further studies would be needed to better characterize the effect on glucose uptake. Similarly, the monocarboxylate transporters MCT-1 and MCT4 were not modified in response to anti-PD1 (Figure 4B,C). However, we observed a decreasing trend for LDH-A staining (*p* = 0.06, n = 4), showing a potential role in the ratio of decreased lactate to pyruvate (Figure 4D). Further, the flux through a pathway is not dictated by expression levels of the enzymes but rather by their activity, emphasizing the need for in vivo real-time dynamic flux measurements. In addition, the lack of change in these proteins at early time points excludes them as early markers of response to anti-PD1 therapy.

## 3. Discussion

Biomarkers are needed to identify patients that resist early treatment in the course of their therapy, allowing them to be moved to potential alternative therapies and/or to avoid any unnecessary side effects. Therefore, identifying patients who will respond to immunotherapy constitutes an important unmet clinical need. In particular, there is currently no consensus to determine which advanced melanoma patient will benefit from immune checkpoint blockers such as PD-L1 expression, mismatch repair deficiency, and tumor infiltrating lymphocytes (TILs), which all failed to be predictors in the clinical setting. Therefore, our aim was to evaluate the relevance of hyperpolarized (HP) ^13^C-pyruvate and ^13^C-MRS fluxomic after U-^13^C-glucose injection as early-response markers to anti-PD1 therapy in syngeneic melanoma xenografts. The YUMMER1.7 melanoma cell line was previously shown to be sensitive to the immune checkpoint blockade, and is, therefore, relevant to the evaluation of the immune checkpoint blockade (ICB) and anti-tumor responses [38]. In the syngeneic YUMMER1.7 model, we confirmed that the PD1 blockade, at a dose of 10 mg/kg, significantly delayed tumor growth in comparison with the isotype control group on day 9 following treatment initiation, potentially induced by the combined effect of an increasing trend in cell death, as assessed using the apoptotic marker caspase-3, and a trend to a decrease in tumor cell proliferation, as shown by the proliferation marker Ki-67, although still with a lack of statistical significance for the sample sizes under study (n = 5–7/group). Nevertheless, our data are in accordance with recent studies showing similar effects in YUMMER1.7 melanomas [39]. In our study, the immunofluorescence analysis of immune cell populations assessed after two cycles of immunotherapy showed an increasing trend in the T cell population at the edge of the tumor and in the macrophage populations in the same area. However, the small sample size in our metabolic imaging study, along with a large variability in the distribution of IHC data, did not allow for significance to be reached. Of note, the immune cell populations were not affected yet after one cycle of immunotherapy. Nevertheless, this observation correlates with a previous study where it has been shown that YUMMER1.7 tumors are infiltrated with CD4 and CD8 T cells [37], and that the infiltration of T cells and macrophages is significant on day nine after implantation in regressing tumors in response to immunotherapy [38]. 

Although ^18^F-FDG PET is a standard clinical metabolic imaging approach for the diagnosis of numerous cancer types, it shows some limitations in the scope of therapy monitoring. In particular, ^18^F-FDG PET has not yet been validated thus far as a robust response marker of the response of advanced melanoma patients to immune checkpoint blockers. This might be due to the fact that ^18^FDG PET only provides information about the glucose uptake and cannot assess downstream metabolites that are important to understanding cancer metabolism, contrarily to ^13^C-glucose tracing and hyperpolarized ^13^C pyruvate MRS methods. In this context, DNP was recently translated into the clinical setting to assess tumor metabolism in human tumors, with a first study of metabolic imaging using HP ^13^C pyruvate reported in prostate cancer patients in 2013 [39], followed by numerous (ongoing) clinical trials. For instance, Brindle et al. showed that the early response to PI3K inhibition could be detected using metabolic imaging with hyperpolarized ^13^C pyruvate, and the assessment of treatment response in breast cancer patients could help to identify patients that benefit from PI3Ka inhibition and design drug combinations to counteract resistance [34]. In addition, it has been shown that hyperpolarized ^13^C pyruvate allows for response assessment in breast cancer patients after seven days of chemotherapy, demonstrating prognostic potential [36]. A recent work also illustrated that HP ^13^C MRI could differentiate tumor aggressiveness in renal cell carcinoma [35]. Of note, HP 13C-pyruvate measurements are technically challenging in the clinical setting since the hyperpolarization decreases quite fast and MRS acquisitions have to be performed within few minutes [33].

Immunotherapy has shown a major breakthrough in the treatment of melanoma. Immune checkpoint inhibitors stimulate the adaptive immune system in order to produce anti-tumor responses [40]. Glucose metabolism regulates T-cell activation and functions. For instance, the Warburg phenotype or aerobic glycolysis used to describe the metabolic shift in cancer cells is a key process to sustain the activation and differentiation of T cells. In addition, lactate is described to have an immunosuppressive impact in the TME [41,42]. Therefore, hyperpolarized ^13^C-pyruvate, a probe used to assess the glycolytic shift in tumors by probing the ratio of ^13^C-lactate to ^13^C-pyruvate in vivo, is a relevant biomarker candidate to indicate early response to immunotherapy. This ratio can be influenced by the direct conversion of pyruvate to lactate by the lactate dehydrogenase (LDH-A), which is often upregulated in tumors; however, it is also regulated via the expression and activity of monocarboxylate transporters MCT-1 and MCT-4, depending on the tumor model under study [43,44]. In this study, we assessed HP ^13^C-lactate to ^13^C-pyruvate exchange in YUMMER 1.7 melanoma xenografts. Interestingly, a significant decrease in the ratio of HP ^13^C-lactate to ^13^C-pyruvate was observed in vivo after one cycle of immunotherapy, which was not observed in the isotype control group. A decrease in the lactate-to-pyruvate ratio is likely due to a decrease in the LDH-A expression in the YUMMER 1.7 melanoma tumors in response to anti-PD1 treatment, although the activity of the enzyme has not directly been studied here. MCT-1, which has been described to rate-limit the conversion of HP ^13^C-pyruvate to ^13^C-lactate in some models [45], was not modified in response to anti-PD1 in the YUMMER1.7 melanoma xenografts. Importantly, the change in the lactate-to-pyruvate ratio preceded any significant change in tumor volume in the anti-PD1 and isotype groups that were significantly different after two cycles of immune therapy. These data illustrate the potential of non-invasive real-time monitoring of the metabolic flux to assess responses before any classic (RECIST) clinical criteria are affected. Further studies using models which are resistant to immunotherapy are required to validate the marker in responsive and non-responsive tumors. 

In addition, the observed changes in the metabolic flux reflect a potential crucial role of the tumor microenvironment and the metabolic interactions between tumor cells and immune cells in the sensitivity of tumors to immune checkpoint inhibitors. For instance, it has been shown that tumors can diminish the functions of tumor-infiltrating lymphocytes and that this metabolic competition can lead to T cell hypo-responsiveness. The PD1 expression on T cells inhibits glycolysis and upregulates fatty acid oxidation, leading to impaired energy generation and decreased effector functions [17,46].

Interestingly, the lack of change in metabolite concentrations assessed ex vivo by ^13^C-MRS fluxomic tracing experiments after U-^13^C-glucose injection illustrates that steady-state metabolite concentrations do not necessarily reflect the activity of a metabolic pathway, contrarily to real-time metabolic measurements. Moreover, differences in the pool of metabolites resulting from a dynamic change in pyruvate lactate exchange can be too subtle to result in a significant change in the steady-state levels of metabolites. Finally, for technical purposes, the timing was slightly different between HP ^13^C-pyruvate and ^13^C-glucose fluxomic experiments (day six and day seven); still, both data were acquired after one cycle of immunotherapy. Overall, our data illustrate that the dynamic monitoring of metabolic fluxes using HP ^13^C-pyruvate can present a significant added value to fluxomic experiments [45,47].

Taken together, these results suggest that metabolic imaging techniques could predict responses to immune checkpoint blockades in the clinical setting. In particular, HP ^13^C-pyruvate could be assessed as an early-response maker in melanoma patients treated with immunotherapy. 

## 4. Materials and Methods

### 4.1. Mice and Tumors

Experiments involving animals were undertaken in accordance with the Belgian law concerning the protection and welfare of the animals, and were approved by the UCLouvain Ethical Committee (agreement reference: UCL/2018/MD/021). Six-week-old C57BL-6 male mice were supplied by Janvier Labs and housed in specific pathogen-free (SPF) environment under standard conditions of temperatures around 20–24 °C and humidity between 45 and 65%. All investigators performing in vivo studies successfully completed FELASA C training.

The YUMMER1.7 mouse malignant melanoma cell line was kindly provided by Professor Leucci, KULeuven, and cultured in Dulbecco’s modified Eagle medium (DMEM), supplemented with 10% heat-inactivated fetal bovine serum (GIBCO, Thermo Fisher Scientific, Waltham, MA, USA) [38].

### 4.2. In Vivo Checkpoint Blockade Treatment 

For the in vivo treatment model, 1.5 × 10^6^ YUMMER1.7 cells in 50 µL of PBS were subcutaneously inoculated in the right hind paw of 6-week-old C57BL-6 mice. Cells were harvested by trypsinization and resuspended in PBS (pH 7.4) before animals were injected.

During inoculation, mice were kept under inhalational anesthesia with isoflurane 2% in 2 L/min airflow.

Tumor-bearing mice were injected intraperitoneally (i.p) with PD1 antibody or isotype control antibodies (10 mg/kg) 3 times per week (days0-2-4-7-9-11) for 2 weeks at a size of 300 ± 50 mm³. Antibodies (in vivo MAb rat IgG2a isotype control and in vivo MAb anti-mouse PD-1 (CD279)) were purchased from BioXcell (Lebanon, NH, USA). After the treatments were interrupted, tumor regrowth was longitudinally monitored using an electronic caliper and calculated via the formula: 4/3 × π×(L/2) × (l/2)^2^. The growth delay of melanoma xenografts was calculated as the time taken (in days) to reach a volume of 1000 mm^3^.

### 4.3. Tissue Fixation, Freezing and IHC Stainings

After sacrifice, the individual tumor from each study animal was collected and divided in half. One half of the tumor was immediately snap-frozen in liquid nitrogen and stored at −80 °C until protein extraction. The remaining portion was immediately fixed in 4% paraformaldehyde for 24 h at room temperature. Samples were subsequently transferred into an automated tissue processor and embedded in paraffin. Following deparaffinization, the inactivation of endogenous peroxidases, antigen retrieval in citrate or Tris-EDTA buffer, and non-specific binding blocking, 5 µm sections were incubated overnight at 4° C with the primary antibodies for MCT1 (Proteintech, Rosemont, IL, USA, #20139-1-AP, 1:1000 dilution) and MCT4 (Sigma, Saint Louis, MO, USA, #HPA021451, 1:500 dilution), LDHA (cell signaling, Danvers, MA, USA, #3558, 1/100 dilution), Ki-67 (cell signaling, Danvers, MA, USA, #12202, 1/300 dilution), and caspase-3 (cell signaling, Danvers, MA, USA, #9661, 1/300 dilution). Consequently, sections were incubated at room temperature for 30 min with the Envision anti-rabbit secondary antibody (Dako Agilent, Santa Clara, CA, #K4003) and stained with diaminobenzidine for 5 min (Dako Agilent, Santa Clara, CA, #K3468). Stained slides were then digitalized using a Pannoramic SCANII slide scanner (3DHistech) at X20 magnification and analyzed using Visiopharm software. The quantification algorithm was run in the viable part of the tissue samples to detect the stained area and the analyzed tumor area. A % of the stained area was calculated as the ratio between the stained area and the analyzed tumor area multiplied by 100.

### 4.4. Multiplex Immunofluorescence (mIF)

After deparaffinization, 5 µm tissue sections were processed according to the protocol described by Aboubakar et al. [48]. Endogenous peroxidases were inhibited for 20 min with 3% hydrogen peroxide in methanol. Sections were then subjected to antigen retrieval in the 10 mM citrate buffer (pH 5.7) and to blocking of aspecific antigen-binding sites (Tris buffered saline (TBS) + bovine serum albumin (BSA) 5% + 0.1% Tween20). Anti-F4/80 primary antibodies (cell signaling, Danvers, MA, USA, #70076, 1/800 dilution) were incubated overnight at 4 °C in TBS containing 1% BSA and 0.1% Tween20, and were detected by corresponding horseradish peroxidase (HRP)-conjugated polymer secondary antibodies (Enision anti-rabbit, Dako Agilent, Santa Clara, CA, K4003) for 40 min at room temperature. The HRP was then visualized by tyramide signal amplification (TSA) using AlexaFluor555-conjugated tyramide (Thermo Fisher Scientific, Waltham, MA, USA, #B40955, 1/200 dilution). After the antibodies were stripped with a new citrate buffer incubation step (this step detached antibodies from tissue sections), the same protocol was applied with the anti-CD3 primary antibody (cell signaling, Danvers, MA, USA, #70076, 1/50 dilution) and Alexa Fluor647 tyramide (Thermo Fisher Scientific, Waltham, MA, USA, # B40958, 1/200 dilution). After a washing step in PBS, the nuclei were finally stained with Hoechst 33342 (Thermo Fisher Scientific, Waltham, MA, USA,), diluted in TBS containing 10% BSA and 0.1% Tween 20, washed in TBS containing 0.1% Tween 20, and mounted with Dako fluorescence mounting medium (Dako, Agilent, Santa Clara, CA, USA). Slides were stored at −20 °C until multispectral image acquisition using an Axioscan.z1 (Zeiss, ×20 objective).

### 4.5. Computer-Assisted Quantitative Evaluation of Immunostaining in Whole Tissue Sections

Whole multiplex-stained paraffin sections were quantified using the image analysis tool Author (version 2022.01.4) (Visiopharm, Hørsholm, Denmark). Tissue sections were first automatically delineated at a low digital magnification (×5) using a thresholding classification method. Blur areas, damaged tissue, and artefacts were manually excluded. To evaluate the stained area, pixels stained for each marker were detected at high resolution (×20) using a thresholding classification method. Results were expressed as a percentage of the stained area within the analyzed periphery region.

### 4.6. Hyperpolarized ^13^C-MRS

Imaging experiments were performed at day 0 (pre-treatment) and day 5. For hyperpolarization experiments, 40 µL of [1-^13^C] pyruvic acid (Sigma-Aldrich, Saint Louis, MO, USA) solution, containing 15 mmol/L of trityl radical OXO63 and 2 mmol/L of gadolinium, were hyperpolarized in an Oxford Dynamic Nuclear Polarizer (HyperSense, Oxford, UK) for approximately 45 min at 1.4 K and 3.35 T. The polarized solution was rapidly dissolved in 3 mL of heated buffer containing 100 mg/L of EDTA, 40 mmol/L of HEPES, 30 mmol/L of NaCl, 80 mmol/L of NaOH, and 30 mmol/L of non-HP unlabeled lactate. This solution was quickly injected using a catheter into the tail vein of the mice in the MRI scanner (11.7-Tesla, Bruker, Biospec, NEST Platform, UCLouvain). Mice were scanned using a double-tuned ^1^H-^13^C-surface coil (RAPID Biomedical, Rimpar, Germany), as previously described [49], which was designed for the spectroscopy of subcutaneous tumors [49,50].

^13^C spectra acquisition and HP [1-^13^C] pyruvate infusion processes were started simultaneously. During MR experiments, animals were kept under inhalational anesthesia with isoflurane (2.5% during anesthesia induction, 1–2% during maintenance) in a 2 L/min airflow. The temperature was continuously monitored and kept at 37 ± 1 °C via a warmed water blanket. Anatomic T_2_-weighted images were used to assess the sequence of tumor volume to turbo RARE. After the administration of ±0.2 mL of hyperpolarized pyruvate, ^13^C spectra were acquired using a single-pulse sequence every 3 s for 210 s (70 repetitions), a flip angle of 10°, and an acquisition bandwidth of 50 kHz (10,000 points). 

The ^13^C exchange between HP [1-^13^C] pyruvate and [1-^13^C] lactate was measured as the ratio between the corresponding areas under the curve (AUCs) via a homebuilt Matlab routine (The MathWorks Inc., Portola Valley, CA, USA).

### 4.7. U-^13^Cglucose Administration

Mice were fasted 6 h before the experiment. Blood glucose levels were measured 15 min before, just prior to intraperitoneal U-^13^C-glucose injection (2 mg/g of mouse) (time point 0), and then every 15 min until the sacrifice at the 75th min, whereby the tumor was sampled, according to the protocol published by Yuan and colleagues [51]. Glycemia was measured with a glucose meter (Accu Check, Roche, Switzerland) on blood samples collected from the tail vein. ^13^C-glucose was injected intraperitoneally at day 6. Tumors were resected 75 min after U-^13^C-glucose administration.

### 4.8. Metabolite Extraction and ^13^C-MRS

Tumor biopsies were grinded using a pestle at cryogenic temperature. Following grinding, the fragmented pieces were transferred into a tube and stored at −80° for further analysis (^13^C-MRS experiments).

Polar metabolites were extracted from snap-frozen tumor tissue, as previously described [51,52]. Around 250 mg of tissue was homogenized in ice-cold methanol (4 µL/mg) using a mechanical lyser (TissueLyser II, Qiagen, Hilden, Germany). The homogenate was then centrifuged at 14,000 g at 4 °C for 20 min. The supernatant was collected and centrifuged again with the same parameters, and any debris was eliminated. After the second centrifugation, the extract for each sample was collected in glass NMR tubes. The solvent was completely removed using a vacuum concentrator. The sample was reconstituted in a 600 µL sodium phosphate buffer with 10% deuterium oxide containing 0.75 wt% 3-(trimethylsilyl) propionic-2,2,3,3-d4 acid (TSP) (Sigma-Aldrich, Saint Louis, MO, USA). Then, ^13^C NMR spectra were acquired on a 600 MHz NMR (Bruker, Biospec, Karlsruhe, Germany), equipped with a broadband cryoprobe, as described previously [53]. The acquisition time was 0.8 s with 2048 repetitions, and an interpulse delay of 10 s (1D sequence with inverse-gated decoupling using a 30° flip angle). Spectrum analysis, assignment, and quantification were performed with MestReNova software (version 14.2.0-26256) (Santiago de Compostela, Spain). Metabolites were quantified by peak integration relative to internal standards and were corrected for tumor mass per sample.

### 4.9. Statistical Analysis

Unpaired *t*-test analysis was performed via GraphPad Prism 9.1.2 (software), and *p* < 0.05 was considered significant. The results are represented as means ± SEM.

## Figures and Tables

**Figure 1 ijms-24-02499-f001:**
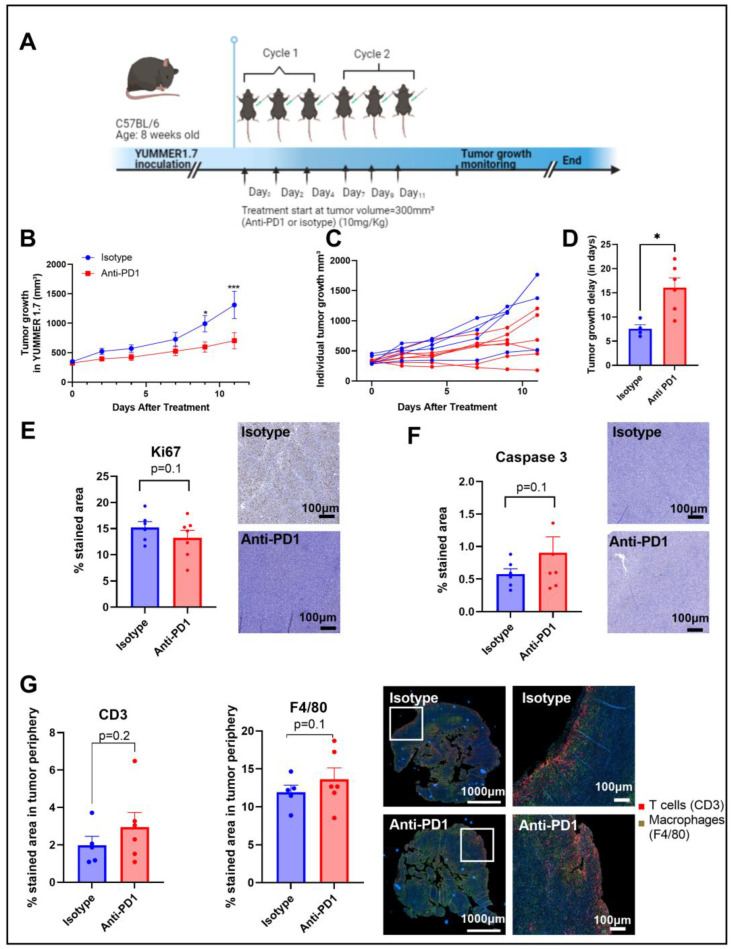
Timeline of the in vivo protocol (**A**). Tumor growth curve, individual tumor growth, and corresponding calculated growth delay as the time (in days) to reach a volume of 1000 mm³ in YUMMER1.7 melanoma xenografts (**B**–**D**). IHC staining for Ki67 and caspase 3 after 2 cycles of IT (**E**,**F**); sample size n = 5–7/group. Illustrations and quantification of T cell (CD3-red) and macrophage (F4/80-green) expressions evaluated by multiplex immunofluorescence (IF) in melanoma tumors after 2 cycles of IT (total tumor area and the tumor periphery area) (**G**). *: *p* ≤ 0.05; the dots correspond to the data points; *** *p* ≤ 0.001.

**Figure 2 ijms-24-02499-f002:**
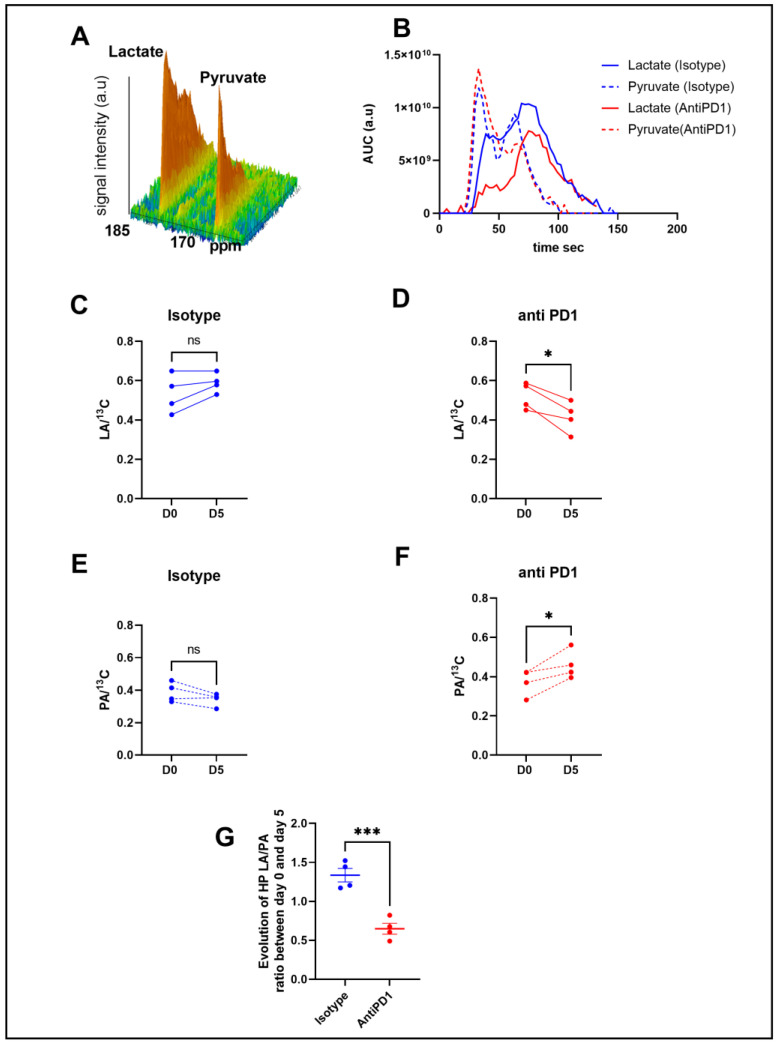
In vivo monitoring of hyperpolarized ^13^C pyruvate–lactate exchange in response to the PD1 blockade. Representative spectra of the ^13^C signal time course obtained from a mouse at baseline (**A**). Representative evolution of the area under the curve of ^13^C pyruvate and ^13^C lactate peaks over time with isotype and anti-PD1 treatment (**B**). Individual changes in calculated ^13^C-lactate/total ^13^C-signals in treated PD1 and isotypes, respectively (**C**,**D**). Individual changes in calculated ^13^C-pyruvate/total ^13^C signals in treated PD1 and isotypes, respectively (**E**,**F**). Evolution of exchange between HP pyruvate and lactate (measured as the ratio of the AUC of [1-^13^C] lactate/the AUC of [ 1-^13^C] pyruvate) in melanoma xenografts after 1 cycle of immunotherapy (**G**). *t*-test n = 4/group. *: *p* ≤ 0.05; *** *p* ≤ 0.001. The dots correspond to the data points.

**Figure 3 ijms-24-02499-f003:**
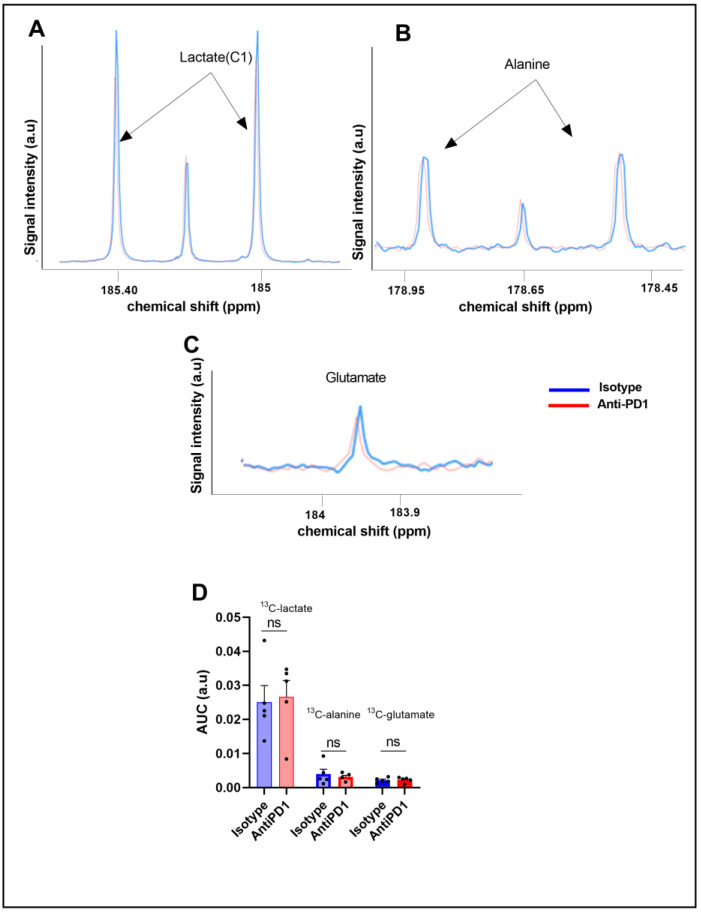
Ex vivo ^13^C metabolic profiling after ^13^C glucose feeding (fluxomic experiments) in response to the PD1 blockade. Representative spectra of lactate C1, alanine, and glutamate issued from the ^13^C glucose metabolism in the control group (blue) and in the anti- PD1-treated group (red) (**A**–**C**). Quantification of the ex vivo ^13^C-MRS detectable metabolites in YUMMER1.7 xenografts, represented by the area under the curve corrected for the internal standard (TSP) and tumor mass arbitrary units (a.u.) n = 5/grp (**D**). The dots correspond to the data points.

**Figure 4 ijms-24-02499-f004:**
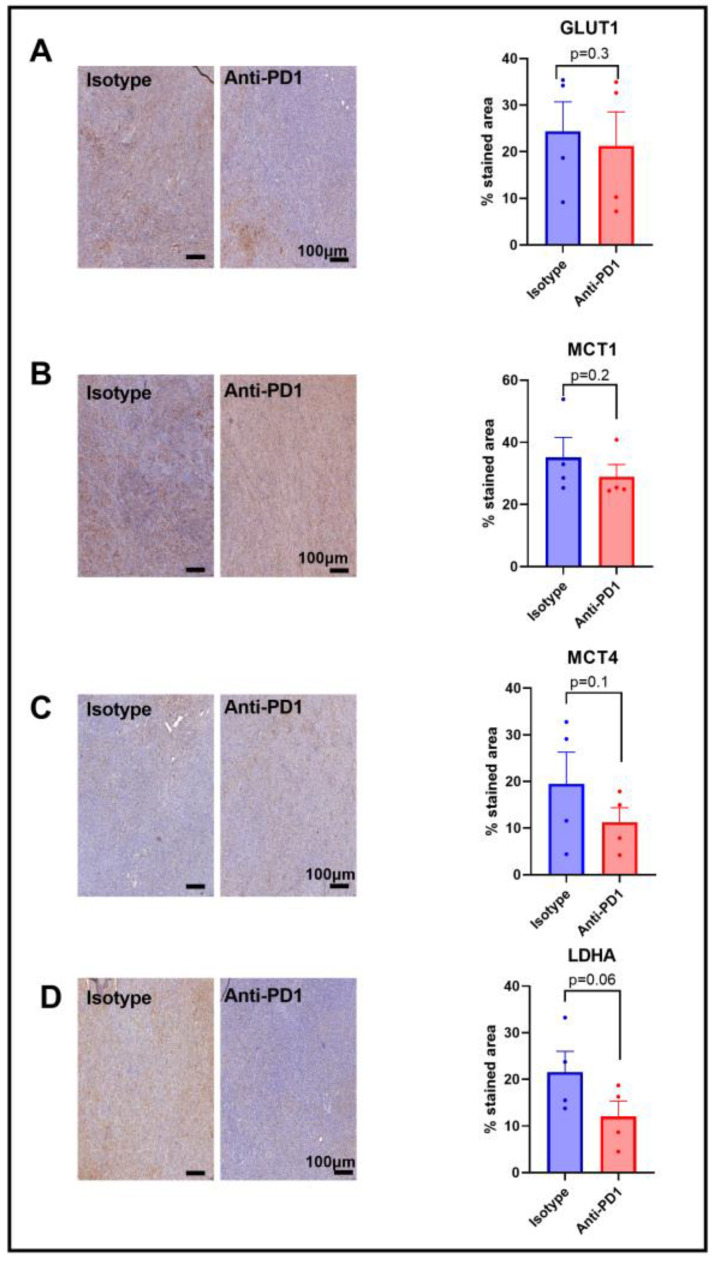
Expression of major glycolytic transporters and LDH-A in response to anti-PD1 in YUMMER1.7 melanoma xenografts. IHC staining for GLUT1 (**A**), MCT1 (**B**), MCT4 (**C**), and LDHA (**D**) obtained after 1 cycle of immunotherapy (isotype control or anti-PD1). Original magnification: 20X; scale bar: 100 µm. Unpaired *t*-test, sample size: n = 4/group. The dots correspond to the data points.

## Data Availability

The data presented in this study are available on request from the corresponding author.

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
