# Peer review of "Hyperpolarized 13C-Pyruvate to Assess Response to Anti-PD1 Immune Checkpoint Inhibition in YUMMER 1.7 Melanoma Xenografts"

_ijms, 2023, doi:10.3390/ijms24032499_

Round 1
Reviewer 1 Report
The authors used metabolic imaging to detect early response to immune checkpoint blockade therapy. The finding of hyperpolarized 13C lactate could be used as a marker to indicate metabolic changes before any significant change in tumor size. However I have several major concerns:
1. How is T cell and macrophage infiltration after one-cycle of PD-1 treatment?
Is it possible the lower lactate/pyruvate ratio is caused by immune cell infiltration, but not tumor cells themselves?
2. Can Hyperpolarized 13C-pyruvate reflect the efficacy of PD1 therapy and overall survival? A different dose should be done.
3. The data quality in Figure 4 is not qualified and is not enough to support the conclusion glucose uptake is not altered, but lactate to pyruvate ratio is decreased in anti-PD1 treated tumor
Minor:
Fig 1F. a lower magnification image should be used to indicate both the edge and the center of tumor.
Fig 4. make sure height and width of the image are adjusted at the same time.
Author Response
We would like to thank you for your report, and for noticing these points. We appreciate your comments. We will find the response to each of your question below.
- How is T cell and macrophage infiltration after one-cycle of PD-1 treatment?
Is it possible the lower lactate/pyruvate ratio is caused by immune cell infiltration, but not tumor cells themselves?
T cell and macrophage infiltration is not significantly after one cycle of anti-PD1 as shown in the figure below. The infiltration is assessed on the totality of the tumor and after one cycle we didn’t observe any significant change. Therefore, this is not likely that the effect on the lac/pyr ratio would be due to immune cells, although not totally excluded. The goal is to identify an in vivo marker of response. This marker could originate from the tumor cells or the tumor microenvironment. In both cases, the marker would be qualified as the goal is to identify early changes in the tumor tissue in its globality. We added this information in the discussion.
(note: see attached pdf to see the data after 1 cycle of IT)
- Can Hyperpolarized 13C-pyruvate reflect the efficacy of PD1 therapy and overall survival? A different dose should be done.
We used hyperpolarized 13C-pyruvate as a biomarker of response to anti-PD1 therapy in the YUMMER1.7 melanoma model. The dose of anti-PD1 was chosen based on the literature and on preliminary experiments. To further validate the relevance of the marker, it would indeed be important to compare this marker in a situation where the response is lower, for example in another model that would be resistant to immunotherapy and to check the marker is not affected in such situation. As an immunoresistant model we use the YUMM1.7 model and experiments are currently ongoing, but are not in the scope of the present proof of concept paper. We added a comment about this point in the discussion part of the manuscript.
- The data quality in Figure 4 is not qualified and is not enough to support the conclusion glucose uptake is not altered, but lactate to pyruvate ratio is decreased in anti-PD1 treated tumor
We indeed did not assess the expression of GLUT-4 and cannot totally exclude an effect on glucose uptake. This has now been adapted in the manuscript. We however suggest that the decrease in the lactate to pyruvate ratio is likely to be due to a decrease in the LDH-A expression in the YUMMER 1.7 melanoma tumors in response to anti-PD1 treatment, although the activity of the enzyme has not been directly studied here. MCT-1, that has been described to be rate-limiting for the HP 13C-pyruvate to 13C-lactate conversion in some models, was not modified in response to anti-PD1 in the YUMMER1.7 melanoma xenografts. Therefore, further studies are needed to definitely conclude about the origin of the change in the lactate to pyruvate ratio. This has been added in the discussion.

Reviewer 2 Report
This paper proposed Metabolic markers such as 13C magnetic resonance spectroscopy of hyperpolarized substrate may bridge this gap, as they allow to assess crucial metabolic fluxes whose alteration is indicative of treatment response or tumour progression. They showed the effect preceded both changes in tumour volume and in tumour oxygenation, thus indicating that the HP lactate/pyruvate ratio may serve as an early marker of response to BRAFi in melanoma. Contrarily to the in vivo settings, BRAFi significantly decreased the HP lactate/pyruvate ratio in vitro, suggesting that such conversion is highly influenced by tumour microenvironment.
The article has good fluency and novelty, and some suggestions for authors.
1. Figure 2b. The time axis is displayed as 0 and 24hrs, and the time axis should be subdivided to let readers understand the change process more clearly.
2. Figure 3 a,b, and c please quantify the unit of the y-axis.
3.Figure 4 b, c in the text y-axis, please reconfirm the % and string standard area middle space problem.
Author Response
We would like to thank you for your feedback and suggestions. We appreciate your comments.
We did the required modifications in the figures. However, regarding figure 2B we assessed our marker at day 0 and day 5 post treatment so we showed just those 2 timepoints on the time axis.
